# Latent Class Analysis of Health Behavior Changes Due to COVID-19 among Middle-Aged Korean Workers

**DOI:** 10.3390/ijerph19031832

**Published:** 2022-02-06

**Authors:** Eun-Hi Choi, Won-Jong Kim, Eun-Mi Baek

**Affiliations:** 1College of Nursing, Eulji University, Uijeongbu-si 11759, Korea; choieh@eulji.ac.kr; 2Department of Preventive Medicine, College of Medicine, Catholic University of Korea, Seoul 06591, Korea

**Keywords:** COVID-19, health behavior, office workers, middle-aged

## Abstract

The purpose of this study was to identify the latent class for changes in health behavior due to COVID-19, reveal the characteristics of participants by type, and identify predictive factors for these types. The participants of this study were office workers between the ages of 40 and 60 and secondary data from the 2020 Community Health Survey of G city was utilized. Latent class analysis was performed on physical activities such as walking and exercise, eating fast food or carbonated drinks, eating delivered food, drinking alcohol, and smoking. Three types of health behavior changes due to COVID-19 were found: (1) decrease in all health behavior type, (2) increase in fast food and delivered food type, and (3) increase in smoking maintenance type. Second, the variables predicting the three types after controlling for general characteristics were health problems, social distancing among the COVID-19 quarantine rules, refraining from going out, and meeting with friends and neighbors and had an impact on COVID-19 life. It is necessary to strengthen non-face-to-face health promotion activities along with quarantine rules for COVID-19. In addition, there is a need for a health management plan for people with non-visible risk factors such as obesity and high blood pressure.

## 1. Introduction

The World Health Organization (WHO) officially declared COVID-19 a “Public Health Emergency of International Concern (PHEIC)” on 30 January 2020, and on March 11, it was declared a “pandemic” [1]. One way to reduce the spread of COVID-19 is to minimize face-to-face contact with others [2]. The U.S. Centers for Disease Control and Prevention (CDC) recommends avoiding situations with risk of infection and maintaining social distance to limit close contact with infected people as much as possible [3]. The Korea Centers for Disease Control and Prevention (KCDC) has emphasized social distancing by wearing masks, prohibiting gatherings, and refraining from going out since 20 January 2020, when the first case of COVID-19 occurred.

With the continuity of the COVID-19 pandemic, people have refrained from using public transportation, and personal daily life has changed, such as refraining from going out and increasing non-face-to-face meetings. The prolongation of the COVID-19 pandemic has also changed individual health behaviors. Naughton et al. [4] stated that dietary changes, such as changes in the intake of fruits and vegetables, and the type and amount of exercise changed, but health behavior deteriorated. Arora and Gray [5] found that there were changes in sleep, dietary behaviors, physical activity, and alcohol consumption, and expressed concerns about mental health.

Countries worldwide reported worsening of people’s mental health such as the occurrence of depression and anxiety due to the prolonged COVID-19 pandemic. In Hong Kong, 25.4% of the people reported that their depression and anxiety were exacerbated by COVID-19, which was related to concerns about COVID-19 infection, lack of masks, and difficulty working from home [6]. Upon conducting a systematic literature review and meta-analysis, it was found that the prevalence of depression increased by 15.97%, anxiety by 23.89%, insomnia by 23.87%, and psychological distress by 13.29% due to COVID-19 [7]. In the case of workers, anxiety, depression, and high work fatigue were found and in the case of families with children, the pandemic also had an effect on support for families of dual-income parents [8].

The UK Health and Safety Executive (HSE) has issued warnings about changes in health behavior and mental health in the case of office workers due to telecommuting, refraining from talking at work, installing partitions between desks, and adjusting the operation of lounge rooms [9]. Similarly, the Ministry of Health and Welfare of South Korea has issued warnings about the dangers of COVID-19 and the consumption of carbonated drinks, delivered food, and convenience food, which can cause nutritional imbalance, among the rules of healthy living [10]. Among office workers, the changes in middle-aged workers are particularly important not only because they are in the peak period of personal and social activities of the human life cycle, but also because physical aging becomes noticeable, and the occurrence of chronic diseases such as metabolic disorders, chronic diseases, and malignant diseases is high. Middle age reduces the resistance to diseases and external stresses due to a decrease in immune function, along with cellular aging, which increases the likelihood of chronic diseases such as hypertension, diabetes, and obesity [11]. Middle age is a period of transition to old age and the question of how to spend this period is important for the quality of life in old age [12]. Since middle-age health is a precursor to old-age health, paying attention to the former can be a way to prepare for the latter.

So far, while there have been some studies looking at changes in health behavior during the COVID-19 pandemic, very few have studied whether there are latent classes for the health behavior change of middle-aged office workers, and what factors characterize the type. Recently, there has been an increasing number of studies regarding health behavior changes through combinations of actions that are moving away from the existing analysis-centered studies of single variables. Among them, Latent Class Analysis is used as a person-centered approach to identify the type of behavior based on individuals’ response patterns [13]. The purpose of this study was to identify the latent classes for the health behavior change of middle-aged office workers in the COVID-19 pandemic, reveal the characteristics of participants by type, and analyze the predictive factors for these types, thereby providing basic data for development of interventions for each type of health behavior change. In this study, middle age is defined as the period between 40 and 60 years of age [14]. The specific objectives were as follows:

First, identify the general characteristics of participants and changes in health behavior due to COVID-19.

Second, typify and name the latent classes for the change in health behavior.

Third, identify the predictive factors for each latent class of the participants’ health behavior change.

## 2. Materials and Methods

### 2.1. Study Design

This was a descriptive study to identify latent classes for changes in health behaviors caused by COVID-19 among middle-aged office workers in Korea and to analyze the influencing factors by type, using secondary data from the 2020 Community Health Survey of G city [15]. G city was chosen because it is the most populous area in South Korea and is representative as it illustrates both urban and rural characteristics. The COVID-19 epidemic in South Korea began in February 2020, and the survey period was from August to October 2020 during the COVID-19 pandemic. 

### 2.2. Participants and Data Collection

The Community Health Survey used in this study is a survey conducted every year since 2008 based on the Korean Community Health Act to identify health behaviors and health conditions, such as presence of diseases in the region, targeting adults aged 19 years or older based on the population living in the area on July 31 of each year. Stratification is done by administrative district and housing type and the sample size is set at 900 per administrative district. For the sample, primary probability proportional phylogenetic extraction is performed by administrative district and housing type and secondary sampling is performed based on the number of households.

In the community survey of this study, a trained surveyor visited the households selected for the samples and conducted the survey through a 1:1 interview using a laptop equipped with a survey program. The data collected were anonymized by deleting information that could identify specific individuals or groups.

This study used community health survey data from August 16 to October 31 conducted in G city in 2020. A total of 41,983 people participated in the 2020 G city Community Health Survey, out of which 16,884 (40.2%) were between the ages of 40 and 60. Among them, those who had no job, were in the army, and those employed in agriculture and fishery were excluded, resulting in 12,629 persons; the 8 persons who checked “non-response” to the variables used in this study were further excluded. Finally, 12,621 responses were analyzed. This study was approved for the use of community health surveys in accordance with the regulations on the procedures for disclosure of raw data from the Korea Centers for Disease Control and Prevention and was approved by the E University Institutional Bioethics Committee (EUIRB2021-071).

### 2.3. Instruments

#### 2.3.1. General Characteristics and Health Problems

The general characteristics of the study participants included age, gender, household type, occupation, and work position. Household types were divided into single-person households, couples, households with parents and children, and other cohabitation households. Occupations were classified into professionals, administrators, office workers, sales and service workers, technicians, and simple laborers. Employee positions were divided into employers and owner operators, wage workers, and unpaid family workers.

Participants’ health problems included hypertension, diabetes, and depression. These three health problems were significantly managed by health centers in South Korea; however, due to the COVID-19 pandemic, they were deprioritized. Therefore, this health problem prioritization should be considered as an important change in health behavior in South Korea. Diagnosis of hypertension was decided by the response “yes” or “no” to the question “Have you ever been diagnosed with hypertension by a doctor”? Diagnosis of diabetes was decided by the response “yes” or “no” to the question “Have you ever been diagnosed with diabetes by a doctor”? For depression, depression scores range from 0 to 27 points by adding up the scores for 9 PHQ-9 items. PHQ-9 score of 1–4 reflects normality, 5–9 mild depression, 10–14 moderate depression, 15–19 moderately severe depression, and 20–27 severe depression. In this study, the summed score was used for depression.

#### 2.3.2. Implementation of COVID-19 Quarantine Rules and Changes in Daily Life

Among the several quarantine rules for COVID-19, social distancing and refraining from going out were selected since they affect interpersonal relationships. On social distancing, participants were asked “Have you practiced maintaining a 2 m (at least 1 m) healthy distance between people (in religious facilities, restaurants, movie theaters, shops, fitness centers, wedding venues, etc.) during the past week”? and had to answer from “Certainly”, “Yes”, “No”, and “Not applicable”. On refrainment from going out, participants chose from “Certainly”, “Yes”, “No”, and “Not applicable” to the question “Have you practiced refraining from going out, gatherings, and events for the past week”?

The number of encounters with friends and neighbors due to changes in daily life due to COVID-19, and the life score affected by COVID-19 were selected. Participants answered from “Increased”, “Similar as before”, “Decreased”, and “Not applicable” in relation to “How has the number of meetings with friends or neighbors changed due to the COVID-19 epidemic compared to before”? As for the life score affected by COVID-19, for the question “If your daily life before the outbreak of COVID-19 is 100 points, and the state of complete interruption is 0, how many points can you give your current daily life status? The scores were classified into 0–20 points, 21–40 points, 41–60 points, 61–80 points, and 81 points or more.

#### 2.3.3. Changes in Health Behavior due to COVID-19

Changes in health behavior due to COVID-19 include physical activity such as walking and exercise, eating fast food or consuming carbonated drinks, eating delivered food, drinking alcohol, and smoking. For each item, participants responded to the question “Does each item change due to the COVID-19 outbreak compared to before?” with “Not applicable” (0 points), “Decreased” (1 point), “Similar as before” (2 points), “Increased” (3 points).

### 2.4. Data Analysis 

Data were analyzed using Mplus 8.5 (Muthen & Muthen, Los Angeles, CA, USA) and SPSS 26.0 (IBM SPSS Statistics, NYC, U.S.) in the following order according to the purpose of the study.

First, the frequency, percentage, mean, and standard deviation of participants’ general characteristics, COVID-19 quarantine rules and daily life changes, and health behavior changes due to COVID-19 were calculated.

Second, latent class analysis was performed on five health behaviors caused by COVID-19. For the fitness for latent class selection, Akaike information criterion (AIC), Bayesian information criterion (BIC), sample size adjusted BIC (sa BIC), Lo–Mendell likelihood ratio test (LMR), and bootstrap likelihood ratio test (BLRT) were used.

Third, multinomial logistic regression analysis was performed to identify factors predicting health behavior change by type of latent class.

## 3. Results

### 3.1. General Characteristics of Study Participants, Characteristics of Health Problems

The average age of the participants was 49.5 years, and the gender distribution was 56.1% men and 43.9% women. For the type of household, single-person households accounted for 8.2%, couples were 13.3%, parents and child households were 66.1%, and others were 12.4%. For occupations, professionals accounted for 20.9%, administrators were 6.6%, office workers were 18.7%, sales and service workers were 22.1%, technicians were 18.6%, and simple laborers were 13.0%. For work positions, employers and owner operators accounted for 25.6%, wage workers for 72.9%, and unpaid family workers for 1.6%.

For health problems, hypertension accounted for 18.9%, diabetes for 7.1%. Mild depression by PHQ-9 accounted for 9.8% and had an average score of 1.8 (Table 1).

### 3.2. Implementation of COVID-19 Quarantine Rules, Changes in Daily Life, and Changes in Health Behavior of Participants

Among the COVID-19 quarantine rules, 61.9% of the participants answered “certainly” and 29.6% answered “yes” to the implementation of social distancing, and 59.7% answered “certainly” and 29.3% answered “yes” to refraining from going out. As for the changes in daily life due to COVID-19, encounters with friends or neighbors “increased” for 0.1% of the respondents, were “similar as before” for 7.4%, and “decreased” for 88.1%. For the life score affected by COVID-19, which was scored from 0 to 100, 9.3% answered 20 or less, 20.1% from 21 to 40, 40.5% from 41 to 60, 23.6% from 61 to 80, and 6.7% from 61 to 80, giving an average of 53.6 points.

Changes in health behavior due to COVID-19 were “increased” for 6.5%, “similar as before” for 38.6%, and “decreased” for 49.3% with respect to physical activity, while fast food and carbonated drinks consumption “increased” for 17.7%, was “similar as before” for 51.1%, and “decreased” for 8.6%. Delivered food consumption “increased” for 23.0%, was “similar as before” for 35.7%, and “decreased” for 6.8%. Drinking “increased” for 5.5%, was “similar as before” for 34.5%, and “decreased” for 48.8%. Smoking “increased” for 3.0%, was “similar as before” 23.1%, and “decreased” for 5.4% (Table 2).

### 3.3. Types of Latent Classes for Changes in Health Behavior due to COVID-19

Latent classes for health behavior changes due to COVID-19 were explored (Table 3). In group 4, LMR and BLRT were not significant. LMR and BLRT were significant in groups 2 and 4, and group 3 had the smallest AIC, BIC, and saBIC values, and thus it was determined as the final model. In group 3, class 1 was 22.1%, class 2 was 51.9%, and class 3 was 26.1%.

Characteristics and names for each latent layer type are presented in Table 4 and Figure 1. Class 1 had a conditional probability value of 1.5 for physical activity, 0.8 for consumption of fast food and carbonated drinks, 0.2 for delivered food, 0.8 for drinking, and 0.1 for smoking. As overall health behaviors decreased, Class 1 was named “decrease in all health behavior type”. Class 2 had a conditional probability value of 1.4 for physical activity, 2.0 for consumption of fast food and carbonated drinks, 2.5 for delivered food, 1.1 for drinking, and 0.1 for smoking. Since the consumption of fast food, carbonated drinks, and delivered food increased, Class 2 was named “increase in fast food and delivered food type”. Class 3 had a conditional probability value of 1.5 for physical activity, 1.7 for consumption of fast food and carbonated drinks, 1.9 for delivered food, 1.5 for drinking, and 2.1 for smoking. As smoking increased, Class 3 was named “increase in smoking maintenance type”.

### 3.4. Predictive Factors by Type of Latent Class for Changes in Health Behavior due to COVID-19

Multinomial logistic regression analysis was performed by inputting all factors according to the derived latent layer type (Table 5).

The regression model was significant (χ^2^ = 20698.541, *df* = 56, *p* < 0.001), Cox and Snell R2 showed 0.288, and Nagelkerke R2 showed 0.331. For general characteristics, age, gender, household type, occupation, and work position were treated as control variables. When class 1 was compared with class 2 as a reference group, hypertension (OR = 1.153, *p* = 0.023) and diabetes (OR = 0.743, *p* = 0.001) were significantly different. For the COVID-19 quarantine rules, when “not applicable” was kept as the standard for the implementation of social distancing, “certainly” (OR = 1.259, *p* = 0.043), “yes” (OR = 1.466, *p* = 0.001), and “no” (OR = 1.907, *p* < 0.001) groups were significant. The practice of refraining from going out was significant for “certainly” (OR = 1.569, *p* = 0.043) and “yes” (OR = 1.547, *p* < 0.001) when “not applicable” was kept as the standard. Regarding changes in daily life due to COVID-19, those where meeting with friends and neighbors “decreased” (OR = 2.138, *p* < 0.001) and “similar as before” (OR = 2.455, *p* < 0.001) were significant, and the life score affected by COVID-19 was significant for 20 points or less (OR = 1.432, *p* = 0.003), 21–40 points (OR = 1.507, *p* < 0.001), 41–60 points (OR = 1.588, *p* < 0.001), and 61–80 points (OR = 1.362, *p* = 0.002) when 81 points or more was kept as standard.

When class 1 was compared with class 3 as a reference group, and “not applicable” was used as the standard for the implementation of social distancing in the COVID-19 quarantine rules, “yes” (OR = 1.502, *p* = 0.004) and “no” (OR = 1.917, *p* = 0.001) groups were significant. Regarding changes in daily life due to COVID-19, when meetings with friends and neighbors “decreased” (OR = 2.133, *p* < 0.001), “similar” (OR = 4.076, *p* < 0.001), and “increased” (OR = 17.952, *p* < 0.001), and the life score affected by COVID-19 was significant at 20 points or less (OR = 1.385, *p* = 0.020) based on 81 points or more.

When class 2 was compared with class 3 as a reference group, and “not applicable” for refraining from going out and “depression” (OR = 1.030, *p* = 0.001) were uses as standards, “certainly” (OR = 0.746, *p* = 0.002) and “yes” (OR = 0.764, *p* = 0.006) groups were significant. Regarding changes in daily life due to COVID-19, meeting with friends and neighbors were significant for those who replied “similar as before” (OR = 1.660, *p* = 0.001) and “increased” (OR = 7.117, *p* = 0.003). The life score affected by COVID-19, based on 81 points or more as standard, was significant for 21–40 points (OR = 0.709, *p* = 0.002), 41–60 points (OR = 0.713, *p* = 0.001), and 61–80 points (OR = 0.768, *p* = 0.013).

## 4. Discussion

This study identified latent classes for changes in health behavior due to COVID-19 and identified predictive factors for the characteristics and types of participants by type. As a result, first, three types of latent classes of health behavior changes due to COVID-19 were identified. These types were named the “decrease in all health behavior type” (class 1), the “increase in fast food and delivered food type” (class 2), and the “increase in smoking maintenance type” (class 3). Second, the variables that predict the three types include age, gender, household type, occupation, hypertension, diabetes, depression, implementation of social distancing, refraining from going out, changes in meeting with friends or neighbors, and changes in life affected by COVID-19. Among the three groups, the “decrease in all health behavior type” accounted for 22.1% of the total participants and it was characterized by a decrease in physical activity, consumption of fast food and carbonated drinks, delivered food, drinking, and smoking. The characteristics and predictive variables of the types of other groups based on the “decrease” type are discussed as follows.

The latent class that accounts for the largest proportion of the total participants was the “increase in fast food and delivered food type”. It accounted for 51.9% of the total participants. Poelman et al. [16] reported that adults with normal weight used food delivery services more frequently during the COVID-19 pandemic. In a study by Carroll et al. [17], people ate less vegetables and more sweets such as snacks. In this study, as a predictor of this, it increased more in the case of hypertension and decreased in the presence of diabetes. In a previous study, the hypertension group had a relatively low awareness of health care compared to the diabetic group [18]. This is because the diabetic group continuously controls their diet and blood sugar, but the hypertensive group has fewer visible symptoms compared to the diabetic group, therefore, the awareness of chronic disease management is relatively low [16]. In the case of hypertension, if there are no symptoms and there is no disruption to daily life, it is thought that individual isolation increased by better following quarantine rules during the COVID-19 period, and the use of fast food and delivered food increased. Bourassa et al. [19] also noted that those who were overweight or obese ate unhealthier food during the COVID-19 period.

In addition, in this study, the predictive factors were that the group with an increase in fast food consumption and delivered food adhered to quarantine rules well and had fewer or similar number of encounters with friends or neighbors. In a study that measured street movement through GPS, residents who undertook a lot of healthy behaviors such as exercise and walking decreased their movement after the COVID-19 pandemic, because health-conscious people actively followed social distancing guidelines [20]. In the COVID-19 pandemic, people are limiting gatherings and the better these rules of conduct are followed, the more social isolation and loneliness increases [21]. These feelings of isolation and loneliness further cause mental stress in people [22], which in turn causes people to eat a lot of food or eating fast food or delivered food in the form of “food cravings” [23]. In addition, as fewer people meet, boredom and stress occur, which increases the tendency to overeat “convenient foods,” which are usually high in calories [24,25], and affects electronic device use and increased screen time exposure [26].

In the case of middle-aged workers who were studied in this research, the use of fast food and delivered food was found to increase in the class where married couples and their children live together. Other studies also showed that delivered food purchasing behavior of the middle-aged group increased [5], and middle-aged and nuclear family consumers avoided going to the store and purchasing food during the COVID-19 pandemic, and purchasing ingredients and food through delivery increased [27], demonstrating similar results to this study. Delivered food tends to be saltier than home-cooked food because it contains a lot of stored and convenience foods. Among them, an increasing proportion of fast food also affects health [28]. 

Of the total participants in this study, 26.1% were from the same or increased smoking group than before the COVID-19 pandemic. In the United States 30.0% and in Israel 44.0% reported that the amount of smoking increased [29,30]. Conversely, in the same study, some smokers said that they quit smoking, but there was not much increase in the amount of smoking in the latent class of this study. As a predictive factor in this latent class, the number of cases where daily life was reduced to less than 20 points due to COVID-19 increased by 1.385 times, and the number of cases with depression increased by 1.036 times as the score increased. Infectious diseases such as COVID-19 are viewed as disasters, and they limit outside activities and reduce interpersonal relationships, leading to deterioration of physical health and mental problems such as anxiety, depression, and stress [31,32]. Although the causal relationship between depression and smoking is unknown in the existing systematic literature review [7], it can be seen that there is a significant relationship among prolonged COVID-19, mental health, and smoking.

The increase in the number of meetings with friends and neighbors was predicted to be 17.952 times for the “increase in smoking maintenance type”. Existing studies have found that there is no relationship between social support and smoking [33], nor a difference in social support according to race [34]. Considering that meeting with friends and neighbors is a form of social support, our study showed different results from these studies. According to some studies, adult smoking types can be divided into five types: considerate of neighbors type, psychological comfort-seeking type, habitual addiction type, self-regulation type, and social relationship-seeking type [35]. In previous studies, partying, relaxation, and seeking spiritual guidance were said to be related to smoking [36]. It is thought that the social isolation and loneliness caused by COVID-19 was resolved by meeting with friends or neighbors as a medium depending on the type of smoking.

In this study, the daily life that was impacted due to COVID-19 had an effect on the health behavior latent class. All participants experienced a decrease in their daily life activities due to COVID-19, and among them, 9.3% of the population increased to 20% or less. In middle age, as physical aging progresses, changes in appearance, weakening of bodily functions, and changes in vitality and metabolism are experienced. In the current pandemic, unfavorable health symptoms for middle-aged individuals, the main economically active population, is obesity, and obesity is linked to chronic diseases such as hypertension and diabetes [37], and it is easy to transition to disease exacerbation [38]. The percentage of decrease in physical activity was 49.3%. Even in the latent type, physical activity was found to decrease regardless of the type. The combination of physical inactivity and an unhealthy diet can predict health risks in people [39]. In particular, as all attention and resources are focused on responding to COVID-19, interest in health promotion may be relatively neglected.

The limitations of this study are as follows. First, the research design was cross-sectional, therefore, it is difficult to estimate the causal relationship between factors. In addition, because this study used secondary data, there was a limitation in that it was not possible to manage or control various variables affecting health behavior. Second, as the participants from this study were selected from one region, it is difficult to broadly interpret it as a survey because the COVID-19 incidence rate, fatality rate, and the degree of implementation of quarantine rules differ from one region to another. Nevertheless, this study is meaningful in that it confirmed the latent types of changes in health behavior due to COVID-19 and looked at predictive factors.

## 5. Conclusions

In order to identify the latent class for changes in health behavior due to COVID-19 and to identify predictive factors for the characteristics and types of participants by type, a secondary analysis was performed using the 2020 G city Community Health Survey.

Changes in health behavior due to COVID-19 occurred such as physical activity including walking and exercise, eating fast food or carbonated drinks, eating delivered food, drinking alcohol, and smoking. There were three types of latent classes: decrease in all health behavior types, increase in fast food and delivered food type, and increase in smoking maintenance type. The “decrease in all health behavior type” accounted for 22.1% of the total participants, the “increase in fast food and delivered food type” was 51.9%, and the “increase in smoking maintenance type” was 26.1%. Variables predicting the potential class included health problems, social distancing due to the COVID-19 quarantine rules, and refraining from going out, meeting with friends and neighbors, and scores that affected life due to COVID-19.

This study was cross-sectional, and thus results cannot be generalized. Nevertheless, it confirmed the potential types of health behavior changes during the pandemic and explored predictive factors.

Based on the results of this study, it is necessary to strengthen the non-face-to-face program for health promotion activities along with quarantine rules for COVID-19. In addition, there is a need for a health management plan for participants with non-visible risk factors such as obesity and high blood pressure.

This study suggests the following recommendations.

First, along with the increase in the implementation of quarantine rules for COVID-19, indoor exercise and smart health management education should be strengthened.

Second, fast foods and salty foods were the mainstays of the existing delivered food, but it is necessary to develop healthy and diverse food in the local community.

Third, mental health and stress management are necessary because smoking is highly likely to increase due to increased stress due to prolonged COVID-19.

## Figures and Tables

**Figure 1 ijerph-19-01832-f001:**
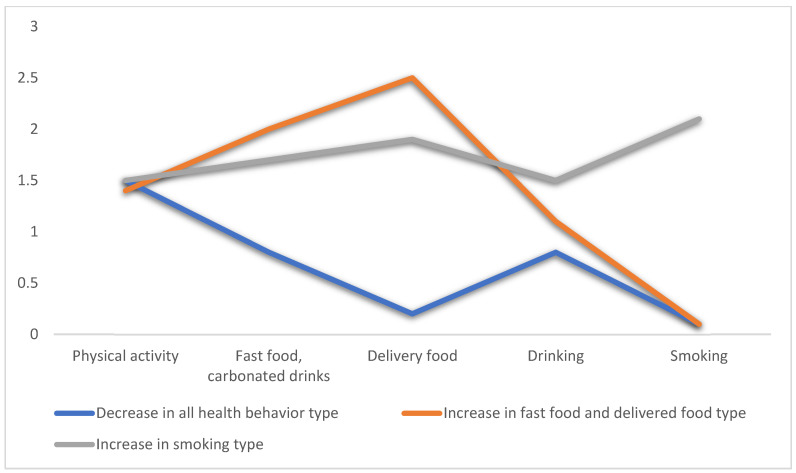
Types of latent classes for changes in health behavior due to COVID-19.

**Table 1 ijerph-19-01832-t001:** General characteristics and health problem characteristics of participants.

Characteristics		Categories	N	%
General characteristics	Age group	40–44	2801	22.2
		45–49	3521	27.9
		50–54	3408	27.0
		55–60	2891	22.9
		M (SD)	49.5 (5.6)	
	Gender	Male	7086	56.1
		Female	5535	43.9
	Household type	One-person households	1032	8.2
		Married couples	1683	13.3
		Parents + children	8347	66.1
		Other cohabitation households	1559	12.4
	Occupation	Professional and related occupations	2643	20.9
		Administrator	834	6.6
		Office worker	2356	18.7
		Sales and service positions	2791	22.1
		Technical job	2351	18.6
		Simple labor	1646	13.0
	Work position	Employers and owner-operators	3225	25.6
		Wage workers	9195	72.9
		Unpaid family workers	201	1.6
Health problems	Hypertension	Yes	2380	18.9
		No	10,241	81.1
	Diabetes	Yes	891	7.1
		No	11,730	92.9
	PHQ-9	Normality	11,142	88.3
		Mild depression	1235	9.8
		Moderate depression	190	1.5
		Moderately severe depression	42	0.3
		Severe depression	12	0.1
		M (SD)	1.8 (2.8)	

**Table 2 ijerph-19-01832-t002:** Participants’ implementation of COVID-19 quarantine rules, changes in daily life, and changes in health behavior.

	Characteristics	Categories	N	%
COVID-19 quarantine rules	Social distancing	Certainly	7812	61.9
		Yes	3741	29.6
		No	482	3.8
		Not applicable	586	4.6
	Refrain from going out	Certainly	7534	59.7
		Yes	3698	29.3
		No	217	1.7
		Not applicable	1172	9.3
COVID-19 daily life changes	Meeting with friends or neighbors	Increased	16	0.1
		Similar as before	936	7.4
		Decreased	11,119	88.1
		Not applicable	550	4.4
	COVID-19 impact life score	≥20	1168	9.3
		21–40	2533	20.1
		41–60	5106	40.5
		61–80	2973	23.6
		≤81	841	6.7
		M (SD)	53.6 (21.2)	
COVID-19 health behavior changes	Physical activity	Increased	819	6.5
		Similar as before	4874	38.6
		Decreased	6225	49.3
		Not applicable	703	5.6
	Fast food, carbonated drinks	Increased	2234	17.7
		Similar as before	6452	51.1
		Decreased	1090	8.6
		Not applicable	2845	22.5
	Delivered food	Increased	4355	23.0
		Similar as before	4502	35.7
		Decreased	858	6.8
		Not applicable	2906	23.0
	Drinking	Increased	699	5.5
		Similar as before	4350	34.5
		Decreased	3631	28.8
		Not applicable	3941	31.2
	Smoking	Increased	381	3.0
		Similar as before	2911	23.1
		Decreased	681	5.4
		Not applicable	8648	68.5

**Table 3 ijerph-19-01832-t003:** Fit indices of latent class analysis and distribution rate of health behavior due to COVID-19.

Number of Groups	AIC	BIC	saBIC	LMR	BLRT	Estimated Probability for Trajectory Group (%)			
						1	2	3	4
1	170,314.599	170,389.031	170,357.252	n/a	n/a	100.0			
2	153,083.774	153,202.864	153,152.017	<0.001	<0.001	73.9	26.1		
3	147,906.303	148,070.051	148,000.137	<0.001	<0.001	22.1	51.9	26.1	
4	144,704.642	144,913.050	144,824.069	0.476	1.000				

Abbreviations: AIC = Akaike information criterion; BIC = Bayesian information criterion; saBIC = sample size adjusted BIC; LMR = Lo–Mendell Likelihood Ratio Test; BLRT = Bootstrap Likelihood Ratio Test.

**Table 4 ijerph-19-01832-t004:** Differences in latent classes for health behaviors due to COVID-19.

Group Indices	Class 1		Class 2		Class 3		F	*p*
	(N = 2784, 22.1%)		(N = 6545, 51.9%)		(N = 3292, 26.1%)			
	M	(SD)	M	(SD)	M	(SD)		
Physical activity	1.5	0.8	1.4	0.7	1.5	0.7	11.357	<0.001
Fast food, carbonated drinks	0.8	0.9	2.0	0.8	1.7	0.9	1842.898	<0.001
Delivered food	0.2	0.4	2.5	0.6	1.9	1.1	10,702.338	<0.001
Drinking	0.8	0.9	1.1	0.9	1.5	0.8	552.132	<0.001
Smoking	0.1	0.3	0.1	0.3	2.1	0.3	66,164.429	<0.001

**Table 5 ijerph-19-01832-t005:** Factors affecting latent class types.

	Characteristics	Categories	Comparison Group (Ref = Class 1)				(Ref = Class 2)	
			Class 2		Class 3		Class 3	
			OR	*p*	OR	*p*	OR	*p*
Health problems	Hypertension (ref = none)	Existence	1.153	0.023	1.049	0.513	0.910	0.125
	Diabetes (ref = none)	Existence	0.743	0.001	0.788	0.020	1.060	0.531
	Depression		1.006	0.503	1.036	0.001	1.030	0.001
COVID-19 quarantine rules	Social distancing	Certainly	1.259	0.043	1.226	0.135	0.974	0.831
	(ref = N/A)	Yes	1.466	0.001	1.502	0.004	1.025	0.848
		No	1.907	<0.001	1.917	0.001	1.005	0.975
	Refrain from going out	Certainly	1.569	<0.001	1.171	0.120	0.746	0.002
	(ref = N/A)	Yes	1.547	<0.001	1.182	0.127	0.764	0.006
		No	1.229	0.306	0.958	0.850	0.780	0.199
COVID-19 daily life changes	Meeting with friends or neighbors	Decreased	2.138	<0.001	2.133	<0.001	0.998	0.986
	(ref = N/A)	Similar as before	2.455	<0.001	4.076	<0.001	1.660	0.001
		Increased	2.522	0.273	17.952	0.001	7.117	0.003
	COVID-19 impact life score	≥20	1.432	0.003	1.385	0.020	0.967	0.790
	(ref ≤ 81)	21–40	1.507	<0.001	1.068	0.593	0.709	0.002
		41–60	1.588	<0.001	1.132	0.270	0.713	0.001
		61–80	1.362	0.002	1.046	0.703	0.768	0.013
	−2 Log Likelihood = 20,698.541 χ^2^ = 4286.357 *df* = 56 *p* < 0.001							
	Cox and Snell R^2^ = 0.288 Nagelkerke R^2^ = 0.331							

Class 1 = Reduced type; Class 2 = Instant and delivered food increase type; Class 3 = Smoking maintenance increase type. Class 1 = Decrease in all health behavior type; Class 2 = Increase in fast food and delivered food type; Class 3 = Increase in smoking maintenance type. Age, gender, household type, occupation, and work position are control variables.

## Data Availability

https://chs.kdca.go.kr/chs/index.do accessed on 29 November 2021.

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
