# Peer review of "Latent Class Analysis of Health Behavior Changes Due to COVID-19 among Middle-Aged Korean Workers"

_ijerph, 2022, doi:10.3390/ijerph19031832_

Round 1

Reviewer 1 Report

The authors identify the latent class for changes in health behavior due to COVID-19, reveal the characteristics of participants by type, and identify predictive factors for these types. The authors found three types of latent classes: decrease in all health behavior types, the variables predicting the three types after controlling for general characteristics were health problems, 18 social distancing among the COVID-19 quarantine rules, refraining from going out, and meeting 19 with friends and neighbors and had an impact on COVID-19 life. I think this paper is original and important for promoting health management plan for people with non-visible risk factors.

I would like to mention the following points I noticed about the paper.

The abstract and title efficiently and clearly indicate the research content.

In the introduction section, the author mentions relevant studies in other countries, but does not mention relevant studies in Korea.  Is this an important issue in Korea? 

The author uses the data of G city for analysis, but does not explain the representativeness of G city in Korea. Can the analysis of the data of G City explain the general situation of Korea? 

The recommendations (lines 365-372) are best placed in the conclusion section.

The conclusions are appropriate based on the results.

Author Response

I would like to extend my deep appreciation to you for review. I thankfully found this round of editing a good chance of improving the paper

Reviewer 2 Report

The article deals with the change in health behavior experienced by workers due to COVID-19. 

In general, it is considered that the article meets the requirements of the journal and its required level of research. It has a suitable research quality, both in its introduction, methodological proposal and results.

I would like to comment on just a couple of issues. First, I believe that the date on which the external survey was carried out, which was at the beginning of the pandemic, should be highlighted and given importance. I think it is important to highlight this because of the variability that there has been in the COVID-19 situation, and the fact that the survey was conducted at the beginning of the pandemic I think is relevant.

On the other hand, I consider that the section "research design" should be expanded a little more. It is mentioned that a latent class analysis is going to be performed, but it is not specified what this analysis consists of and why this analysis is considered suitable to achieve the objectives proposed for the study.

The results are specified in a suitable way.

Author Response

(The authors gave the same response as above.)

Reviewer 3 Report

Overall, I found this to be a well written study on a topic that currently has high interest worldwide so should have large appeal across a wide readership. I have no major potential concerns to raise. My comments for the authors’ consideration are shown below.

  1. Line 48: ‘Conducting’ rather than ‘’conducing’.
  2. Line 66: I recommend adding ‘the’ between ‘to’ and ‘former’.
  3. Line 69: It would be used to briefly define and rationalise the use of latent classes, since this seems to be an important aspect in justifying the need for your study.
  4. Lines 84-87: If other papers have been published relating to the 20202 Community Health Survey of G city it would be useful to cite them here, as readers might be interested in them.
  5. Line 120: It would be useful to briefly justify why these health problems were specifically chosen and why others were not included.
  6. In the UK where I live, delivered food can be from shops/restaurants that deliver cooked fast food such as burgers and pizzas, and shops that deliver your normal weekly groceries such as bread, milk, eggs, etc. I recommend that ‘delivered food’ is defined in the manuscript, as the two examples I give above a clearly different in regards the potential health implications.
  7. Line 155: ‘Were’ rather than ‘was’ (since data are plural).
  8. Line 179: Does the 1.8% refer to the percentage of people in your sample that had depression? If so, I recommend the wording is revised to make this clearer.
  9. Line 300: ‘Feelings’ rather than ‘feeling’.
  10. Lines 328-329: This sentence does not make sense to me. Please consider rewording.
  11. Lines: 337-340: This sentence does not make sense to me. Please consider rewording.

Author Response

(The authors gave the same response as above.)
